# Data-Driven Bi-Directional Lattice Property Customization and Optimization

**DOI:** 10.3390/ma17225599

**Published:** 2024-11-15

**Authors:** Fuyuan Liu, Huizhong Wu, Xiaoteng Wu, Zhouyi Xiang, Songhua Huang, Min Chen

**Affiliations:** 1School of Advanced Technology, Xi’an Jiaotong-Liverpool University, Suzhou 215123, China; fuyuan.liu16@student.xjtlu.edu.cn (F.L.); huizhong.wu22@student.xjtlu.edu.cn (H.W.); xiaoteng.wu22@student.xjtlu.edu.cn (X.W.); zhouyi.xiang22@student.xjtlu.edu.cn (Z.X.); songhua.huang@xjtlu.edu.cn (S.H.); 2School of Engineering, The University of Liverpool, Liverpool L69 7ZX, UK

**Keywords:** generative design, lattice customization, parametric lattice design, machine learning, data-driven lattice exploration

## Abstract

Customizing and optimizing lattice materials poses a challenge to designers. This study proposed a data-driven generative method to customize and optimize lattice material. The method utilizes subdivision modeling to parametrically describe lattice morphologies and skeletons. Next, the homogenization method is employed to analyze elastic moduli for collecting a dataset. Then, a two-tiered machine learning (ML) framework is proposed to predict the elastic modulus for a forward design. The first-tier model employs polynomial regression to estimate relative density, which serves as an additional input feature for the second-tier model. The prediction accuracy of the second-tier model is improved through the additional inputs. The forward and reverse design strategies offer a flexible and accurate means of tailoring lattice properties to meet specific performance requirements. Two case studies demonstrate the practical value of the framework: customizing a lattice material to achieve a desired elastic modulus and optimizing the mechanical performance of lattice materials under relative density constraints. The results show that the prediction accuracy of the elastic modulus using the two-tiered ML model achieved an error of less than 10% compared to finite element analysis, demonstrating the reliability of the proposed approach. Furthermore, the optimization design achieved up to a 25% improvement in mechanical performance compared to conventional lattice configurations under the same relative density constraints. These findings underscore the advantages of combining generative design, machine learning, and genetic algorithms to navigate complex design spaces and achieve enhanced material performance.

## 1. Introduction

The design and optimization of mesoscale lattice materials have become increasingly significant in advanced manufacturing due to their exceptional mechanical and functional properties. These materials, characterized by their periodic cellular architectures, offer high specific stiffness [1], good strength [2], sound energy absorption [3], and even supernatural properties, like a negative Poisson’s ratio [4]. Such attributes make them ideal for high-end industrial applications, including airplane components [5], thermal management systems [6], and medical implants [7]. However, customizing optimized parametric lattice materials to meet specific performance criteria remains a considerable challenge.

Traditional design approaches are predominantly empirical, relying on iterative, manual processes involving geometric modeling, parametric adjustments, and extensive numerical analysis. These methods are time-consuming, costly, and often fail to fully exploit the vast design space available for lattice configurations. The lack of a systematic methodology for exploring and optimizing both the lattice configuration and the morphological features of lattice materials limits innovation and hinders the development of solutions tailored to specific application needs.

Recent advancements in computational design and artificial intelligence (AI) have introduced a generative design paradigm that facilitates the customization and optimization of lattice materials [8,9,10,11]. Generative design replaces manual design processes, enabling the automatic and efficient generation of highly complex and optimized lattice structures. AI technologies, including neural networks [12], generative adversarial networks (GANs) [13], and variational autoencoders (VAEs) [14], have expedited the generation of high-performance lattice materials. For example, Lee et al. used a hybrid neural network and genetic optimization methods combined with Bézier curves to optimize lattice profiles, enhancing the elastic modulus [15].

Furthermore, data-driven design methods enable us to elaborate the relationship between design variables and mechanical properties. This allows for forward property prediction and an inverse lattice material with desired properties. Yayati et al., utilizing a denoising diffusion-based model, accelerated the design process of a TPMS-like lattice unit cell structure with desired mechanical properties outperforming traditional simple cubic cells [16]. Challapalli and Li et al. utilized machine learning to design and optimize lattice configurations, with their optimized lattice cells significantly outperforming octet cells in terms of buckling loads and demonstrating enhanced compressive performance in both experimental and simulation validations [17]. Despite these advancements, a gap remains in fully integrating generative design, parametric modeling, and machine learning for forward-to-inverse lattice material design.

In the generative design paradigm, digital geometry design determines the design space of lattice material. Lattice materials can be considered as a combination of their skeleton and morphology [18]. In terms of skeleton design, the spatial positions of lattice material components and their connections have mainly been studied. For example, Chen et al. enhanced the stiffness, expansibility, and energy absorption capabilities of materials by designing self-similar concave tensile lattice configurations [19]. Ding et al. improved the negative Poisson’s ratio effect of lattice materials by parametrically altering the bending and twisting angles of lattice rods [20]. Guo et al. used a fast Fourier transform-based homogenization method to explore the mechanical properties of mixed materials with multi-lattice configurations based on triply periodic minimal surfaces (TPMSs), enhancing compressive energy absorption and other properties [21]. Rahman et al. demonstrated the potential of lattice materials to improve energy absorption and mechanical efficiency by exploring various rod-based lattice configuration hybrid structures [22]. Compared to skeleton design, strategies for refining lattice morphologies have evolved, ranging from smoothing lattice nodes to parametrically modifying their shapes. Cao et al. found that cross-sectional optimization improves the energy absorption and mechanical performance of the rhombic dodecahedron lattice [23]. Bernard et al. demonstrated that non-circular sections, such as squares or rectangles, markedly improved the resilience and energy absorption of strut-based lattices [24]. Uddin et al. designed I-shaped rod sections that improved the compressive performance of pyramid lattice structures and their resistance to buckling and bending [25]. However, existing research often focuses on either the lattice skeletal configuration or its morphological details in isolation, which restricts the potential for holistic optimization.

To address these challenges, this research proposes a data-driven bi-directional framework that synergizes generative design, machine learning, and optimization algorithms for advanced mesoscale lattice material design. The core objective is to develop a unified approach that simultaneously optimizes both the skeleton and morphology of lattice structures, enhancing their customization and performance across various applications. Parametric SubD modeling is utilized for the detailed digital representation of the lattice structure. A small sample dataset of mechanical properties is collected using the homogenization method, which simplifies the complex lattice structure into an equivalent homogeneous material for efficient analysis. To enhance predictive accuracy, a two-tiered machine learning framework is proposed. The first tier uses polynomial regression to estimate relative density, which is then used as an input feature for the second tier, a Random Forest model that predicts the elastic modulus. A genetic algorithm is employed to customize and optimize lattice designs that meet the desired mechanical specifications and restrictions.

The effectiveness of the proposed approach is validated through numerical simulations and case studies, demonstrating its capability to produce optimized lattice structures that satisfy or surpass desired performance criteria. By integrating generative design, machine learning, and optimization into a cohesive framework, this research provides a comprehensive solution for customizing optimized parametric lattice materials at the mesoscale.

## 2. Design Methodology

The proposed generative design strategy synergizes parametric design, machine learning techniques, and genetic algorithms to allow for the customization of high-performance lattice material. This approach comprises two main phases: dataset collection and inverse–forward lattice exploration of the lattice frame (depicted in Figure 1).

In the first part, the initial lattice skeleton and morphology are digitally defined by a set of geometric parameters $\left[P_{1},P_{2},P_{3},\ldots ,P_{n}\right]$, and subdivision (SubD) modeling is used to construct organic-shaped lattice units. Subsequently, a series of representative volume elements (RVEs) for lattice material is generated by sampling combinations of geometric parameters. The homogenization method evaluates their corresponding mechanical properties, such as Yong’s modulus, Shear modulus, and Poisson’s ratio, resulting in an expansive dataset that maps geometric parameters to mechanical properties.

In the second part, the collective data are used to train machine learning (ML) models to establish the relationship between mechanical properties and geometric parameters. Driven by a genetic algorithm (GA), both forward and inverse designs of lattice materials are implemented: forward design enables performance optimization, while inverse design allows for the customization of lattice materials to achieve specific target properties.

### 2.1. Parametric Lattice Skeleton and Morphology Based on SubD Modeling

The methodology begins with the digital characterization of the lattice skeleton and morphology, as seen in Figure 2. Firstly, the skeleton of a lattice unit is parametrically described; then, the mesh framework for the morphology design of the lattice material is defined. Then, subdivision (SubD) modeling is used to construct smooth, continuous surfaces. This method uses parametric subdivision surfaces to produce high-quality, smoothing lattice material.

In this process, the Catmull–Clark algorithm is a highly effective approach for lattice modeling, as it works well with both triangular and rectangular meshes. This method iteratively calculates new vertex points by averaging the coordinates of the original vertices from the initial mesh structure, as seen in Figure 3. For the modeling process, the Rhino7-Grasshopper^TM^ SubD component is used, which has an internal implementation of the Catmull–Clark subdivision algorithm. Since this component automatically manages the parameters of subdivision algorithms, users are unable access and edit the specific parameters. But it allows one to influence the final subdivision result by controlling the initial control mesh characteristics.

As seen in Figure 2, the process begins by constructing a lattice unit on a skeletal framework, with each lattice strut wrapped in quadrilateral mesh. During the modeling phase, the strut nodes and lattice material profiles are parameterized. Subsequently, the meshes are subdivided into smaller sections based on the topology of a mesh-based lattice unit, forming lattice units with organic shapes. As subdivision iterations increase, the model becomes progressively smoother, as seen in Figure 4.

The use of parametric subdivision technology not only extends the diversity of lattice material but also significantly minimizes stress concentration [27]. By utilizing Rhino7-Grasshopper^TM^ SubD component, we obtained high-quality lattice models, converted into .x_t and .stl format for additive manufacturing and finite element analysis, respectively.

### 2.2. Numerical Analysis Based on Homogenization Method

Following the modeling phase, the homogenization method is applied to evaluate the mechanical properties of the parametrically designed lattice material. This method reduces computational load by treating the complex lattice structure as an equivalent homogeneous material, enabling the efficient analysis of macroscopic properties [28].

The application of the homogenization method involves two key steps: the definition of an appropriate Representative Volume Element (RVE) and its numerical assessment under periodic boundary conditions. As illustrated in Figure 5a,b, a cubic RVE can be composed of a 3 × 3 × 3 lattice unit. This selection ensures stable macroscale mechanical behavior, minimizing the size effect. This also accelerates the collection of the elastic modulus of all samples.

To achieve material homogeneity, periodic boundary conditions were established on the RVE, as illustrated in Figure 5c. The faces of RVE on the *X*-*Y*-, *Y*-*Z*-, and *X*-*Z*-axis planes were defined as master planes, while their opposing, parallel faces were classified as slave planes. A set of three constraint equations was established on the element nodes in the master and slave planes to impose periodic boundary conditions at the corresponding locations. The displacement of the RVE is obtained by imposing a force in the Z-axis direction, as detailed in Equation (1).
(1)$${\left.u_{z}\right|}_{z={RVE}_{size}}-{\left.u_{z}\right|}_{z=0}=\delta_{z}={RVE}_{size}\times \varepsilon_{z}^{0}$$

Here, $u_{z}$ represents the displacement, $\delta_{z}$ denotes the deformation in the Z-axis direction, and $\varepsilon_{z}^{0}$ stands for the normal strain in the Z-axis direction, respectively. ${RVE}_{size}$ is the length of lattice material.

The elastic modulus of the lattice RVE was calculated using Equation (2). During the numerical analysis, the homogenized elastic modulus was incorporated into the ML model as a predicted target.
(2)$$E_{z}=\frac{\sigma_{z}}{\varepsilon_{z}}=\frac{F_{z}/A_{xy}}{\delta_{z}/l_{z}}$$

Here, $E_{z}$ is the Young’s modulus in the z-direction, $F_{z}$ is the load in the z-direction, $A_{xy}$ is the area used by the load to calculate stress $\sigma_{z}$, $\delta_{z}$ is the displacement in the *Z*-axis direction of the structure, and $l_{z}$ is the RVE height, used to calculate strain $\varepsilon_{z}$.

### 2.3. Bi-Directional Lattice Customization and Evaluation

The customization of high-performance lattice material involves the implementation of a two-tiered machine learning (ML) model alongside a genetic algorithm (GA). The first tier of the ML model utilizes polynomial regression to predict the relative density (*RD*) of the lattice based on its geometric features, where the *RD* is the ratio of the actual volume of the lattice to the size of its cell space. Such predictions not only increase the number of input features for the second-tier ML model, but they also reduce the need for lattice volume calculations in traditional CAD software. The second tier, employing a Random Forest algorithm, utilizes the *RD* and geometric parameters to predict the homogenized elastic modulus. Simultaneously, the GA operates in a reverse design framework to identify the optimal geometric parameters that achieve the desired mechanical properties. These steps collectively enhance the forward and reverse engineering capabilities of lattice design, facilitating a more precise and efficient exploration and customization of lattice materials.

#### 2.3.1. Forward Mechanical Property Prediction Method

In forward design, we present a two-layer machine learning (ML) framework for lattice materials to accurately predict their elastic modulus. The first layer of the ML model of the framework enhances the prediction accuracy by extending the input features. In addition to geometrical features, *RD* is also considered as it has a significant impact on the mechanical properties of lattice materials. It is worth noting that there is a strict constraint between the *RD* and geometrical parameters, which means that *RD* is not independently selectable when geometrical parameters are given. Typically, in CAD software, the *RD* is calculated based on the input geometric parameters, but this approach increases the computational cost of the prediction.

To solve this problem, the first layer of the ML model predicts the *RD* directly using the geometric features as inputs, thus reducing the dependency on CAD calculations and significantly reducing the computational burden. In this step, a polynomial regression approach is used to predict the *RD* due to the complex interactions between geometric features that jointly affect the *RD*. This approach effectively captures the complex relationships between geometric features and provides higher prediction accuracy.

Let the geometric features X = ($G_{1}$, $G_{2}$, $G_{3}$,…, $G_{n}$) be the vector of geometric features. The three-degree polynomial regression model for predicting *RD* can be expressed as Equation (3):(3)$$RD=\beta_{0}+\sum_{i=1}^{n}\beta_{i}X_{i}+\sum_{i=1}^{n}\sum_{j=i}^{n}\beta_{ij}X_{i}X_{j}+\cdots +\epsilon$$
where $\beta_{0}$ is the intercept; $\beta_{i}$ and $\beta_{ij}$ are the coefficients for the linear and quadratic terms, respectively. $X_{i}$ and $X_{j}$ are the geometric features involved in X. ϵ is the error term.

By incorporating *RD* as a predicted feature, we enhance the dataset used for subsequent ML models, ultimately improving the overall accuracy of mechanical property predictions. This approach allows for efficient and accurate *RD* estimation, reducing the reliance on computationally intensive CAD software.

The second-tier ML model is Random Forest. The RD is compensated through the polynomial regression as an effective input feature. Hereafter, the Random Forest model is applied to predict the homogenized elastic modulus. This approach ensures both the precision of ML predictions and the efficiency of the design process. Random Forest was chosen due to its robustness and ability to handle complex non-linear relationships between geometric features and the elastic modulus.

Let *E* denote the homogenized elastic modulus. The input features for the Random Forest model include Z = ($G_{1}$, $G_{2}$, $G_{3}$,…, $G_{n}$, RD). The Random Forest model can be described as Equation (4).
(4)$$E=RF\left(Z\right)$$

A single decision tree, *T*, in the Random Forest predicts the homogenized elastic modulus E based on input features *X* in Equation (5).
(5)$$T\left(Z\right)=\sum_{i=1}^{L}w_{i}I\left(X\in R_{i}\right)$$
where L is the number of leaves in the decision tree. $w_{i}$ is the predicted value (homogenized elastic modulus) for leaf i. I(·) is the indicator function, which is 1 if Z falls into region $R_{i}$ and 0 otherwise.

A Random Forest consists of an ensemble of *m* decision trees. Each tree is built on a different bootstrap sample from the training data and a random subset of features. The prediction of the Random Forest is the average prediction of all decision trees in Equation (6).
(6)$$E_{RF}\left(Z\right)=\frac{1}{m}\sum_{j=1}^{m}T_{j}\left(Z\right)$$
where $E_{RF}(Z)$ is the predicted homogenized elastic modulus from the Random Forest. $T_{j}(Z)$ is the prediction of the j-th decision tree for input feature Z. Given the input features, each decision tree $T_{j}$ in the Random Forest provides a prediction $T_{j}(Z)$. The final prediction, $E_{RF}\left(Z\right),$ is the average of all individual tree predictions, as described in Equation (7).
(7)$$E_{RF}\left(X\right)=\frac{1}{m}\sum_{j=1}^{m}T_{j}\left(RD,G_{1},G_{2},G_{3},\ldots ,G_{n}\right)$$

The mean absolute error (MAE), Root Mean Square Error (RMSE), and the coefficient of determination (R^2^) are employed to evaluate the predictive performance of the models concerning lattice materials’ relative density and equivalent elastic modulus in Equations (8)–(10).
(8)$$\mathrm{MAE}\left(y,\hat{y}\right)=\frac{1}{m}\sum_{i=1}^{m}\left|y_{i}-{\hat{y}}_{i}\right|$$
(9)$$\mathrm{RMSE}\left(y,\hat{y}\right)=\sqrt{\frac{1}{m}\sum_{i=1}^{m}{\left(y_{i}-{\hat{y}}_{i}\right)}^{2}}$$
(10)$$R^{2}\left(y,\hat{y}\right)=1-\frac{\sum_{i=1}^{m}{\left(y_{i}-{\hat{y}}_{i}\right)}^{2}}{\sum_{i=1}^{m}{\left(y_{i}-\bar{y}\right)}^{2}}$$
where $y_{i}$ is the computational value of the equivalent elastic modulus and ${\hat{y}}_{i}$ is the predicted value. m is the number of samples and y is the predicted mean response where the equation is $y=\frac{1}{m}\sum_{i=1}^{m}y_{i}$. The MAE is used to evaluate the closeness between the predicted results and the real dataset. The RMSE reflects the deviation between the actual value and the predicted value.

#### 2.3.2. Backward Lattice Customization Design

In the reverse lattice design phase, a genetic algorithm (GA) is employed to identify optimal discrete parameters that meet the objectives of mechanical properties under the constraints between *RD* and geometric parameters.

The geometric parameters of the lattice materials and *RD* need to be encoded as individuals in the algorithm, as seen in Equation (11).
(11)$$V_{Individual}=\left[G_{1},G_{2},G_{3},\ldots ,G_{n}\right]$$

The optimization objective is to minimize the absolute deviation between the predicted elastic modulus and the target value, while also minimizing the *RD*. The objective function is formulated to balance the accuracy of $E_{z}$ and minimization of *RD*, as defined as Equation (12). During the optimization process, some physical and design constraints need to be satisfied to ensure the generated parameter combinations are reasonable, subject to the polynomial constraint expressed in Equation (3).
(12)$$\begin{array}{c} \min f\left(V_{Individual}\right)=w_{1}\cdot \left|E_{z}^{\mathrm{pred}}-E_{z}^{\mathrm{target}}\right|+w_{2}\cdot RD\\ \left\{\begin{array}{c} E_{z}^{\mathrm{pred}}=f_{\mathrm{RF}}\left(V_{Individual},f_{\mathrm{PR}}\left(V_{Individual}\right)\right)\\ G_{1}^{min}<G_{1}<G_{1}^{max}\\ G_{2}^{min}<G_{2}<G_{2}^{max}\\ G_{3}^{min}<G_{3}<G_{3}^{max}\\ \ldots \\ G_{n}^{min}<G_{n}<G_{n}^{max} \end{array}\right. \end{array}$$
where $w_{1}$ and $w_{2}$ are weighting factors that adjust the trade-off between the elastic modulus and relative density during optimization. $E_{z}^{\mathrm{pred}}$ is the ML prediction value for the mechanical performance based on lattice geometric parameters, and $E_{z}^{\mathrm{target}}$ is the desired performance value. $f_{\mathrm{RF}}$ denotes the Random Forest model in which $V_{Individual}$ and *RD* are regarded as inputs, and $f_{\mathrm{PR}}$ denotes the polynomial regression model applied to predict *RD* by $V_{Individual}$.

We used Tournament Selection to select individuals based on their fitness values. We applied multi-point crossover methods to combine parts of the genes from two individuals to generate new individuals. We randomly selected certain genes in individuals and mutated them with a certain probability. Through the above genetic operations, a new generation of the population was generated. The process of fitness evaluation, selection, crossover, and mutation was repeated until the termination condition was met. The specific hyperparameters of the genetic algorithm can be determined based on sensitive analysis in the following cases.

## 3. Case Study: Property Customization

A lattice unit with a nested cube was selected as case study to demonstrate the validation of the proposed method, as seen in Figure 6.

Five geometric parameters were defined to describe the lattice shape as shown in Figure 7a. Four parameters were used to define the lattice morphology: $M_{inn}$ defines the size of the inner nodes; $M_{out}$ defines the size of the outer nodes, $M_{strut}$ describes the radius of the strut size, and $M_{smooth}$ determines the smoothness of the lattice struts. In terms of configurations, the lattice is formed by a cube nested with a small cube; eight vertexes are connected to form a lattice skeleton. $T_{size}$ determines the size of the cubes of lattice materials. This parametric mesh lattice model underwent a transformation into a smoothed lattice model through the Rhino 7-Grasshopper^®^ SubD component, as shown in Figure 7b,c.

To validate the efficacy and advantages of the proposed lattice design approach, a comparison analysis was conducted using conventional parametric lattice designs as benchmarks. The comparison focused on the mechanical properties, particularly the elastic modulus, of the designed lattice materials. As illustrated in Figure 8, *L* defines the box size inside a lattice unit, and *D* denotes the diameter of the lattice struts. Under the same relative density, the elastic moduli of both SubD parametric lattice and regular parametric lattice are presented.

Figure 8 shows that the lattice materials designed using our methodology demonstrated a significantly higher upper boundary of elastic modulus compared to conventional lattices under equivalent mass conditions. This indicates superior material utilization and performance.

The Latin Hypercube Sampling (LHS) method was used to sample the geometric parameter combinations to ensure uniformity of sampling. The specific sampling ranges for each parameter are detailed in Table 1. Considering the minimum resolution of Formlab SLA devices, each parameter was rounded to two decimal places. This clearly determined a global design space encompassing 21 × 21 × 11 × 101 × 6 design possibilities. A total of 280 sets of geometric parameters were extracted through LHS. The Rhino 7-Grasshopper^®^ Anemone plug-in facilitated the automatic generation of lattice models.

A cubic RVE consisting of 3 × 3 × 3 parametric lattice cells (shown in Figure 7b) with a side length of 18 mm was chosen. This choice not only ensures stable macroscale mechanical behavior and minimizes size effects but also reduces the computational burden when calculating the elastic modulus of all samples. FormLab White resin was chosen to fabricate all lattice materials. Its mechanical properties are: a density of ρ = 1.10 g/mm^3^; a Young’s modulus of Es = 2.51 GPa; and a Poisson’s ratio of ν = 0.23 according to previous testing [29]. The RVE was treated as orthogonally isotropic, and a displacement (0.09 mm) was applied to the *Z*-axis direction to obtain the support reaction force and calculate the modulus of elasticity under periodic boundary conditions, as detailed in Equations (1) and (2). The numerical evaluation of all lattice samples was conducted through batch processing in ANSYS Mechanical APDL.

After collecting the dataset, it was essential to address the presence of outliers in the dataset to enhance the model’s accuracy and robustness. We defined outliers based on the statistical properties of the target variable (elastic modulus $E_{c}$). The 99th percentile was the threshold for outlier detection. Any data point where $E_{c}$ exceeded this threshold was regarded as an outlier and removed from the dataset since software may generate a low-quality RVE model, which can lead to an extremely high elastic modulus during numerical evaluation. The filtered data samps are presented in Figure 9b. Then, correlation analyses were conducted, as seen in Figure 9a–d. The analysis of the first-tier data presented a strong correlation between $M_{\mathrm{inn}}$ and *RD*, while other variables also showed a high correlation with *RD*, except for $M_{\mathrm{smooth}}$, as shown in Figure 9a. Additionally, there were no obvious relations among geometric variables. In the second-tier dataset, a similar trend was observed with *RD* and $E_{c}$, where *RD* showed a strong correlation with $E_{c}$, as shown in Figure 9c. The importance of all input features is shown in Figure 9d.

In the ML training process, the dataset was split into a training set and a validation set (80% of the data for training and the remaining 20% for validation) using a random split technique. To prevent bias in the sequence of the data for model training, the data were randomly shuffled before splitting.

By modulating the order of the polynomial regression, we found that the model fits best when the order is 4. To optimize the performance of the Random Forest regression model, we performed hyperparameter tuning using a grid search approach. This method exhaustively searches through a specified parameter grid to find the combination of hyperparameters that yields the best performance based on five-fold cross-validation results. The hyperparameters and their respective search ranges were specified as follows:Number of Estimators ($n_{\mathrm{estimators}}$): [100, 200, 300].Maximum Depth (${Depth}_{max}$): [None, 10, 20, 30].Minimum Samples Split (${Samples\_split}_{min}$): [2, 5, 10].Minimum Samples Leaf (${Samples\_split}_{min}$): [1, 2, 4].Maximum Features (${Features}_{max}$): [‘auto’, ‘sqrt’, ‘log2’].

After conducting the grid search, the best combination of hyperparameters was found to be $n_{\mathrm{estimators}}$ = 300; ${Depth}_{max}$ = 10; ${Samples\_split}_{min}$ = 2; ${Samples\_split}_{min}$ = 1; and ${Features}_{max}$ = ‘sqrt’.

Additionally, to determine the average performance index, five-fold cross-validation (CV) was used to randomly split and train many times. This increased the robustness of the model performance evaluation and lessened the random effects that could be caused by a single data split. The result, illustrated in Table 2**,** reveals the robust predictive capabilities of the ML model.

Additionally, to demonstrate the reliability of this ML model, a robustness analysis was conducted under different noise levels. Gaussian noise ranging from 1%, 2%, 5% to 10% was imported into each of the input features to simulate measurement errors. The results of the study showed that the model maintained strong robustness and stability under varying noise levels, as seen in Figure 10. The R^2^ scores fluctuated slightly in the noise interval from 1.0% to 10.0%, indicating that the model was able to retain a high level of goodness-of-fit in the presence of noise disturbances. This was particularly significant, as the model still effectively explained the main variability in the data at higher noise levels. The MSE and MAE increased gradually with rising noise, but the increase was more moderate, further validating the model’s adaptability under medium-to-high noise conditions. Despite the rise in error due to noise, the model showed a degree of noise tolerance at noise levels of 5.0% and above, which is crucial for property prediction.

In the reverse design phase, a genetic algorithm was used to identify lattice materials that satisfied tailored properties while maintaining minimum relative density, as defined in Equation (13). The range of geometric parameters is defined in Table 1.
(13)$$\begin{array}{c} \min f\left(V_{Individual}\right)=w_{1}\cdot \left|E_{c}^{\mathrm{pred}}-E_{c}^{\mathrm{target}}\right|+w_{2}\cdot RD\\ \left\{\begin{array}{c} E_{c}^{\mathrm{pred}}=f_{\mathrm{RF}}\left(V_{Individual},f_{\mathrm{PR}}\left(V_{Individual}\right)\right)\\ 0.30<M_{\mathrm{inn}}\;<0.50\\ 0.30<M_{out}<0.50\\ 0.5<M_{\mathrm{off}}\;<1.5\\ 1.0<M_{\mathrm{ss}}\;<10.0\\ 1.0<T_{\mathrm{bs}}\;<1.5 \end{array}\right. \end{array}$$
where both $w_{1}$ and $w_{2}$ determine the weight of the elastic modulus and relative density, respectively. $E_{c}^{\mathrm{pred}}$ is the model’s prediction for the mechanical performance based on the input parameters, and $E_{c}^{\mathrm{target}}$ is the desired performance value. $f_{\mathrm{RF}}$ denotes the Random Forest model in which $V_{Individual}$ and *RD* are regarded as inputs, and $f_{\mathrm{PR}}$ denotes the polynomial regression model applied to predict the *RD* by $V_{Individual}$. The population size for each generation was set to 100, with 50 iterations. Further experiment validation was achieved through the customization of four distinct lattice materials, targeting elastic modulus values within the range of 80 to 120 MPa while ensuring the lowest possible relative density. Before running the genetic algorithm, a sensitive analysis was conducted. Two weights values, $w_{1}$ and $w_{2}$, were explored firstly. We first set the target value of the modulus of elasticity retrieved by the genetic algorithm to 160 MPa, set a series of weighting parameters, and observed that their effects on the best-fit values (as shown in Table 3) are varied. The weights satisfy ($w_{1}$ + $w_{2}$ = 1), which ensures a constraint on the total weights.

Figure 11 illustrates the sensitivity analysis of the weighting parameters $w_{1}$ and $w_{2}$ to the best fitness value. As $w_{1}$ increases and $w_{2}$ decreases, the optimal fitness value gradually increases. The results show that higher weighted $w_{2}$ values help to reduce the fitness value and thus reduce the relative density more efficiently, whereas higher $w_{1}$ values focus the optimization on modal matching but tend to lead to higher fitness values. Hence, $w_{1}$ and $w_{2}$ were set as 0.1 and 0.9, respectively. After determining the sensitivity of $w_{1}$ and $w_{2}$, we further explored the sensitivity of the genetic algorithm parameters.

Population size (${Pop}_{size}$): [50, 100, 150].Iteration number ($N_{gen}$): [15, 20].Crossover probability (${Cx}_{pb}$): [0.4, 0.5].Mutation probability (${Mut}_{pb}$): [0.1, 0.2, 0.3].Tournament size (${Tourn}_{size}$):): [2, 3].

Figure 12 shows that the performance of the genetic algorithm is significantly affected by the number of iteration generations and the probability of variation. Increasing the number of iteration generations ($N_{gen}$ = 20) can effectively improve the convergence of the algorithm so that the algorithm reaches the optimal fitness value of 3.5416 many times and avoids the sub-optimal solution problem caused by stopping too early. A higher variance probability (${Mut}_{pb}$ = 0.2 or 0.3) shows advantages in expanding the search space and avoiding local optimums, significantly improving the quality of the solution. In addition, tournament size (${Tourn}_{size}$ = 3) helps balance exploration and exploitation with a moderate increase in selection pressure, resulting in more stable results. Comparatively, crossover probability (${Cx}_{pb}$ = 0.4 or 0.5) and population size (${Pop}_{size}$ = 50 to 150) have less impact on the algorithm results and can be flexibly adjusted according to resources. But ${Pop}_{size}$ is set 100 to balance the computational cost with the performance of the algorithm. This combination is expected to improve the quality of the solution and ensure the stable convergence of the algorithm at the same time. Overall, appropriately increasing the number of iteration generations and choosing a higher variance probability are key measures to improve the performance of the algorithm. Based on the results of the analysis, the recommended final parameter settings are as follows:$$w_{1}=0.1,w_{2}=0.9,\;N_{gen}=20,\;{Mut}_{pb}=0.3,\;{Tourn}_{size}=3,\;{Cx}_{pb}=0.4\;{Pop}_{size}\;=100$$

In the experimental section, we attempted to customize four optimal lattice materials with target elastic moduli ranging from 80 to 120 MPa and the lowest relative density. We selected 3 × 3 × 3 lattice units and conducted compression tests on a Sansi Zongheng universal testing machine. Each lattice sample was produced five times using stereolithography (SLA) on a FormLab3 printer. Post processing included cleaning the sample with a Xiaomei ultrasonic cleaner in 99.8% isopropanol for 10 min to eliminate any residual resin and ensure the cleanliness and integrity of the sample. The lattices were subsequently cured in a FormLab UV curing machine and gradually heated to 60 °C over 60 min. During the printing and cleaning process, damaged lattices were excluded. The final selection of three lattice structures from each group for compression testing was primarily influenced by two factors. Firstly, the Formlabs equipment exhibited printing delamination issues, which affected the integrity of some lattice materials. Secondly, the removal of support structures caused damage to certain lattices, rendering them unsuitable for testing. Thus, the three chosen lattices were those that remained intact and retained their geometric features after these challenges. Table 4 analyze the actual results of physical experiment $E_{c}^{\exp}$ and record stress-strain curves of each sample.

To describe the relative error (RE) between the predicted relative volume (${RD}_{pred}$) and the actual relative volume from CAD software (${RD}_{CAD}$), the following equation is used, as shown in Equation (14):(14)$$RE=\frac{{RD}_{pred}-{RD}_{CAD}}{{RD}_{CAD}}\times 100\%$$

In assessing relative density, ${RD}_{pred}$ showed variability depending on the target elastic modulus $E_{c}^{\mathrm{target}}$. As seen relative density error curves in Table 5, ${RD}_{pred}$ showed higher errors at lower $E_{target}$ values, diminished errors at moderate $E_{target}$ values, and increased errors once again at higher $E_{target}$ values. Conversely, the polynomial regression-predicted ${RD}_{pred}$ displayed smaller positive errors at lower desired properties, with errors slightly increasing, then shifting to negative at moderate $E_{target}$ values and decreasing again at higher values, as seen elastic modulus error curves in Table 5. Notably, at $E_{target}$ = 100 MPa, the error in ${RD}_{pred}$ reached −1.18%. This indicated a more stable prediction from polynomial regression as confirmed by data in Table 5.

In assessing relative density, ${RD}_{pred}$ exhibited variability depending on the target elastic modulus ($E_{target}$). As depicted in Table 5, ${RD}_{pred}$ showed higher errors at lower $E_{target}$ values, diminished errors at moderate $E_{target}$ values, and increased errors once again at higher $E_{target}$ values. Conversely, the polynomial regression-predicted ${RD}_{pred}$ displayed smaller positive errors at lower $E_{target}$ values, with errors slightly increasing, then shifting to negative at moderate $E_{target}$ values and decreasing again at higher values. Notably, at $E_{target}$ = 100 MPa, the error in ${RD}_{pred}$ reached −1.18%. This indicates a more stable prediction from polynomial regression. The larger variance in ${RD}_{pred}$ might be attributed to a higher delta setting as shown in Equation (11).

In terms of customizing the elastic modulus, the relative errors between the predicted elastic modulus ($E_{pred}$) and $E_{target}$ were minimal. This precision proves the capability of GA to effectively navigate the parameter space defined by the two-tiered ML model to optimize lattice designs. For the errors between $E_{sim}$ and $E_{pred}$, the results suggest that the prediction capability of ML is not stable but can be referenced as the fluctuation range is from 0.75% to 8.55%. $E_{pred}$ exhibits larger errors at higher $E_{target}$ values, decreasing gradually with increasing $E_{target}$ and stabilizing at lower $E_{target}$ values. It is noteworthy that the deviation trend exhibits high agreement between the prediction error of the elastic modulus and *RD*. As the error of ${RD}_{pred}$ increases, the error of $E_{pred}$ also amplifies, consistent with the result of feature importance analysis. The *RD* error of polynomial regression can further expand the elastic modulus error of Random Forest regression. Each of the four designed lattice materials met or exceeded the targeted mechanical properties, thereby validating the predictive accuracy and practical applicability of our design methodology.

Despite these challenges, all four designed lattice materials successfully met or surpassed their targeted mechanical properties, affirming the robustness and practical applicability of our design method. The GA was instrumental in determining the optimal discrete parameters within the polynomial constraints of relative density and geometric parameters. Its application highlights the algorithm’s efficiency in managing the intricate design space of lattice materials, leading to highly precise material property achievements.

## 4. Case Study: Performance Optimization

The detailed parametric characterization of lattice materials significantly facilitates the exploration of the morphological features of lattice material. This approach not only makes the exploration of lattice morphology more flexible but also broadens the spectrum of achievable mechanical performance for lattice materials. Initially, a body-centered cubic (BCC) lattice unit was modeled using quadrilateral meshes, with each strut having hexagonal cross-sections. Four geometric parameters were defined to describe the lattice morphology, as shown in Figure 13a. S1 is the distance parameter which constructed points and from which the strut midpoint extended along both sides. S2 and S3 specify the strut radius at both S1 points and the midpoint of the strut, respectively. N1 represents the lattice node size. This parametric mesh lattice model was transformed into a smoothed one using the Rhino 7-Grasshopper^®^ SubD component, as shown in Figure 13b. The use of subdivision modeling technology ensures an organic morphology while significantly reducing stress concentrations.

To efficiently explore the vast potential design space and collect geometric features, Latin Hypercube Sampling (LHS) was used. This statistical approach helps to sample four geometric parameters, ensuring uniform coverage across the range of each variable. The specific sampling ranges for each parameter are detailed in Table 6.

Considering the minimum resolution of our additive manufacturing equipment, each parameter should be rounded to two decimal places. This clearly determined a global design space encompassing 11 × 36 × 36 × 36 design possibilities. A total of 238 sets of geometric parameters were sampled, generated, and evaluated following the procedure described in Section 3. The deformed 3 × 3 × 3 RVE and its von Mises stress distribution under the compressive load are illustrated in Figure 14.

The distribution of this lattice sampling is presented in Figure 15a, and then the parameters’ sensitivity is demonstrated in Figure 15b. Relative density plays a vital role in determining the elastic modulus, while S1, the distance parameter, has less influence on the mechanical properties of the lattice structure. Node size and strut radii (N1, S2, and S3) have a significant contribution to the lattice’s mechanical performance.

The dataset was collected and pre-processed in the same way as in Section 3. This dataset was split into 80% for training and 20% for validation, and five-fold cross-validation was applied to assess the models’ effectiveness after grid searching hyperparameters of the Random Forest model. The predictive outcomes, illustrated in Table 7**,** highlight the strong predictive performance of the ML models.

The outcome of the two-tiered ML model was the production of *RD* and Young’s modulus forecasts for a global design space of parametric lattice materials. This dataset allowed designers to efficiently search and filter lattice materials within the Rhino 7-Grasshopper^®^ environment, pinpointing options that meet specific mechanical performance criteria and constraints across 11 × 36 × 36 × 36 possible lattice design configurations, as shown in Figure 15a.

The performance of the machine learning models was validated using the metrics provided in Table 7. Linear regression achieved a CV MAE of 0.0049 and a CV R^2^ of 0.98187, indicating high predictive accuracy for relative density. Random Forest regression, used in the second tier, resulted in an MAE of 4.5046 and an R^2^ of 0.91912, showing good predictive capability for Young’s modulus despite the increased complexity of the task. These results highlight the robust performance of the ML models in capturing the non-linear relationships between lattice parameters and mechanical properties, even when handling complex design configurations.

In the experimental phase, we customized four optimal lattice materials with a relative density in the range of 0.155 to 0.160 and maximum elastic modulus. The 6 × 6 × 6 lattice units were selected for compressive tests on a SanSiZongHeng universal testing machine. In the optimization process, we selected the first four parameter combinations with the best fitness at the time of convergence of the genetic algorithm. These parameter combinations represent the design solutions with the best performance under the current optimization conditions. Three lattice materials were produced for each parameter combination and subjected to the same post-processing methods as in Section 3. They were subsequently compressed, and their compression modulus was averaged. A conventional BCC lattice served as the benchmark for comparing mechanical performance, as detailed in Table 8.

Table 8 presents a detailed comparison between the traditional BCC lattice material and four customized lattice materials (C1, C2, C3, C4) and shows the stress-strain curves of all samples during the compression testing. Images show that the FormLab3 SLA printer accurately captured the geometric features of the digital model, with all samples maintaining a *RD* strictly within the target range of 0.15–0.16. The compressive modulus (Es) metrics revealed the mechanical performance enhancements achieved through the proposed filter strategy. Specifically, C1 exhibited the highest compressive modulus at 11.27 MPa, an increase of about 25.6% over the conventional BCC lattice’s 8.97 MPa. C4 also showed an elastic modulus 21.5% higher than the conventional lattice. Notably, C1 and C4 exhibited closely matched values across parameters, indicating similar geometric features that likely influence their mechanical performance. This significant improvement highlights the benefits of lattice customization in enhancing material stiffness under compressive loads.

## 5. Discussion

The data-driven bi-directional framework proposed in this study demonstrates significant potential in lattice material design optimization, successfully integrating generative design, machine learning (ML), and genetic algorithms (GA) to achieve highly accurate performance prediction and lattice material optimizations. However, we also recognize that there are some limitations and assumptions in this study, which need to be discussed in depth to clarify the scope of applicability of the methodology and the direction of improvement.

Firstly, we assume that the geometric parameters and relative density can accurately predict the elastic modulus. This simplification may not fully capture the effect of print orientation on the mechanical behavior of the lattice structure in additive manufacturing, especially when the lattice structure is complex and has significant anisotropy. Although relative density is an important factor influencing the elastic modulus, ignoring the effects of microgeometry and local stress concentrations on the mechanical properties of the material during the printing process may lead to deviations in the predicted results from the actual properties. Secondly, we adopted the homogenization technique to simplify the complex lattice structure and thus improve computational efficiency. Secondly, we adopted the homogenization technique to simplify the complex lattice structure and thus improve computational efficiency. However, the method may introduce inaccuracies in approximating the mechanical properties of heterogeneous materials, especially for lattices with complex topologies and significant size effects. Such simplifications may not adequately reflect the subtle interactions within the lattice, leading to discrepancies between numerical predictions and physical results. In addition, this study assumes that the material exhibits linear elastic behavior during loading and does not consider non-linear behaviors such as plastic deformation, viscoelastic effects, or damage evolution.

During customization and optimization, the two-tiered machine learning models (polynomial regression and Random Forest) rely on the quality and diversity of the training data. Since the training data are mainly from numerical simulations, poor meshing accuracy during finite element analysis may increase the deviation in numerical analysis properties from the actual mechanical properties of lattice materials. The prediction accuracy may decrease. Meanwhile, the performance of genetic algorithms is highly sensitive to their parameters (e.g., population size, crossover probability, variance probability, and selection strategy). Although we performed parameter sensitivity analyses to identify appropriate settings, the selected parameters may not guarantee a globally optimal solution. In addition, this study does not consider constraints in the manufacturing process such as minimum feature sizes, tolerances, and defects that may occur during processes such as additive manufacturing. These factors may significantly affect the performance and feasibility of the lattice structure in actual production, and ignoring them may lead to design solutions that are difficult to implement in reality.

The above uncertainties are the main reason leading to deviations between the actual performance and predicted results. The limited number of experimental tests we conducted may not be sufficient to fully assess the accuracy of the model, and experimental errors and randomness of the samples may affect the reliability of the results.

To address these limitations, future research should focus on improving model accuracy and generalization by expanding the dataset to include more diverse lattice geometries, cell types, and material behaviors. Enhancing homogenization techniques and adopting multi-scale modeling can provide more accurate predictions by capturing both overall structural behavior and local effects. Further, integrating fabrication constraints into the optimization framework will ensure designed lattice structures are not only performance-optimized but also highly manufacturable. Comprehensive experimental validation is crucial, involving increased sample sizes and a wider range of tests to validate predicted performances. Establishing a feedback loop to iteratively improve ML models and optimization algorithms using experimental results will enhance model accuracy and robustness. Finally, developing adaptive optimization strategies, such as adaptive genetic algorithms that dynamically adjust parameters during optimization, can enhance convergence to global optimal solutions.

## 6. Conclusions

This research proposes a bi-directional lattice design method that leverages generative design and machine learning to facilitate both the forward optimization and reverse design of lattice materials. The approach addresses the inefficiencies and trial-and-error limitations of traditional methods by providing a robust framework for advanced lattice material customization and optimization. The proposed method achieves notable improvements in accuracy, efficiency, and design space exploration by integrating parametric design for both the lattice skeleton and morphology, a genetic algorithm, and a two-tiered machine learning framework. The principal outcomes of this research can be summarized as follows:The implementation of SubD modeling to describe both the skeleton and morphology of lattice materials broadens the mechanical property space and significantly improves mechanical performance compared to regular lattice structures.The two-tiered machine learning framework enables deeper insights into the complex interrelations within lattice morphology, skeleton parameters, and relative density, leading to substantial advancements in the ability to predict and control material properties. This approach achieves target properties more accurately than traditional optimization methods that rely solely on geometric parameters.The parametric lattice design and bi-directional customization capability support a broader exploration of the design space, allowing for both the property-driven generation of structures and the discovery of novel configurations. This flexibility makes the approach especially suitable for applications that demand customized lattice structures.The optimized lattice materials demonstrate over a 25% improvement in the elastic modulus compared to regular geometric lattices, affirming the effectiveness of the approach in enhancing lattice performance.

The proposed method demonstrates a balanced enhancement in accuracy, efficiency, and flexibility in design exploration, making it well suited for advanced lattice structure optimization compared to traditional state-of-the-art approaches. By simplifying the customization and optimization processes, it opens new avenues for designing high-performance lattice structures and broadens the potential for lattice materials in specialized applications.

## Figures and Tables

**Figure 1 materials-17-05599-f001:**
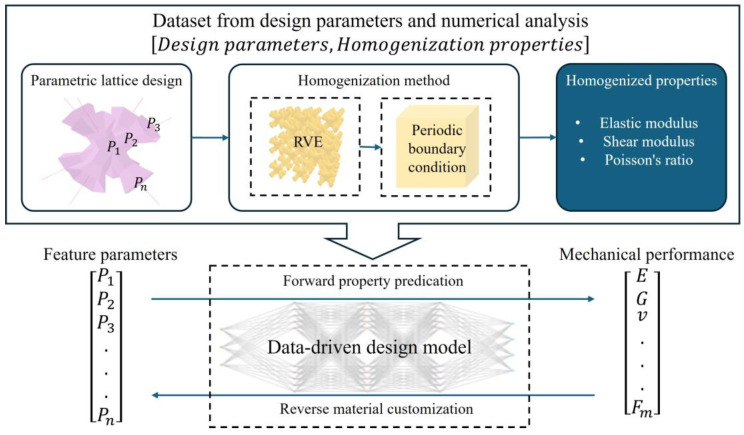
Design framework.

**Figure 2 materials-17-05599-f002:**
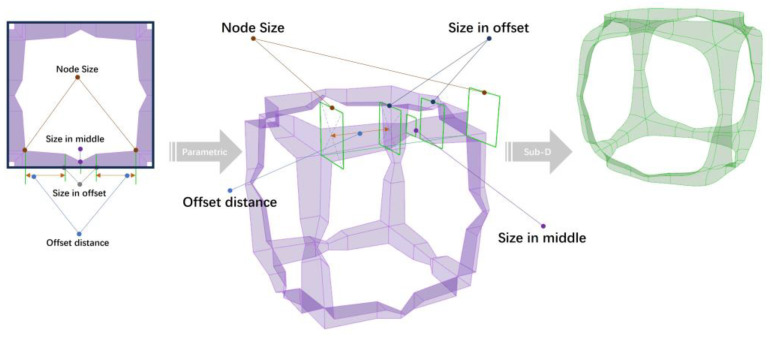
Parametric modeling and SubD lattice units.

**Figure 3 materials-17-05599-f003:**
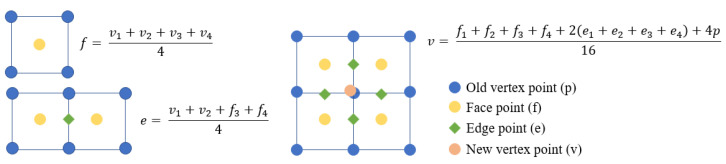
Catmull–Clark algorithm explanation [26] (Copyright © 2024, Liu et al., under exclusive license to Springer Nature Singapore Pte. Ltd.).

**Figure 4 materials-17-05599-f004:**
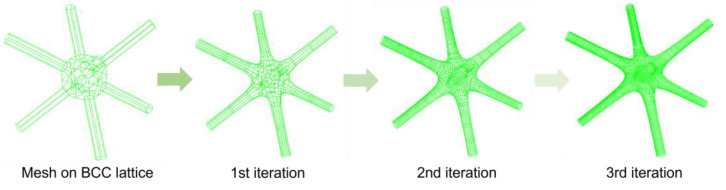
Catmull–Clark SubD modeling method for smoothing lattice unit.

**Figure 5 materials-17-05599-f005:**
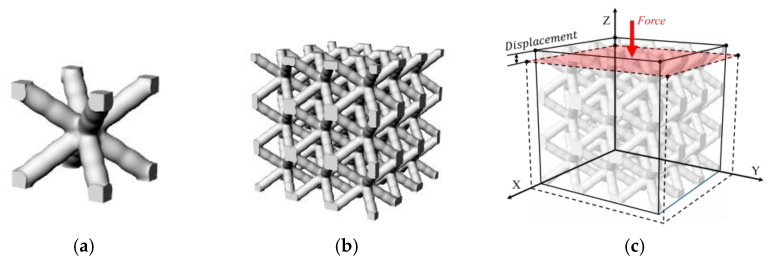
Homogenization approach: (**a**) lattice unit; (**b**) 3 × 3 × 3 RVE; (**c**) a homogenized analysis with periodic boundary conditions and a *Z*-axis load.

**Figure 6 materials-17-05599-f006:**
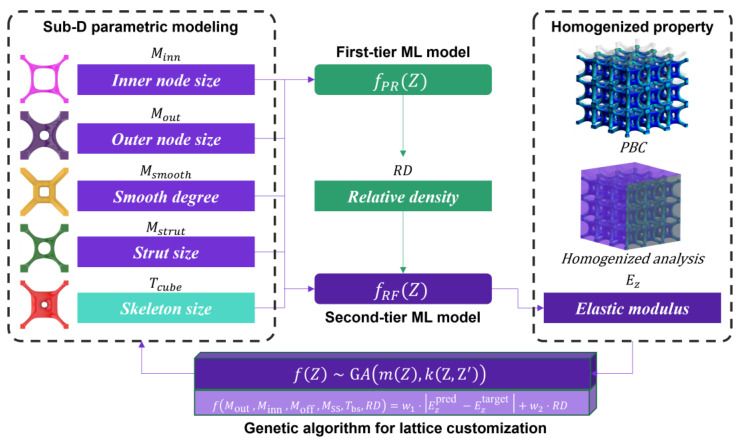
A practical design frame for bi-directional customization.

**Figure 7 materials-17-05599-f007:**
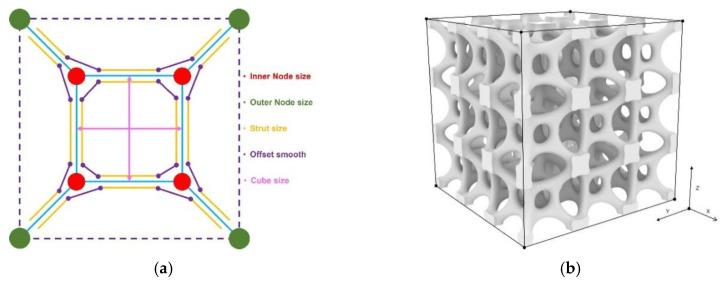
Parametric lattice design based on subdivision modeling: (**a**) parametrizing four morphology design variables, including inner node size, outer node size, strut size, and smooth level, and one topology variable, the size of a nested box; (**b**) a parametric 3×3×3 lattice unit; (**c**) the influence of each parameter to geometric features of a lattice unit.

**Figure 8 materials-17-05599-f008:**
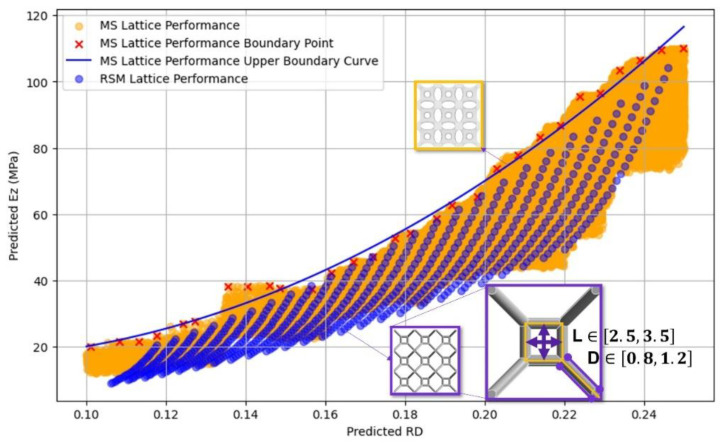
Mechanical performance difference between the proposed lattice and regular lattice material.

**Figure 9 materials-17-05599-f009:**
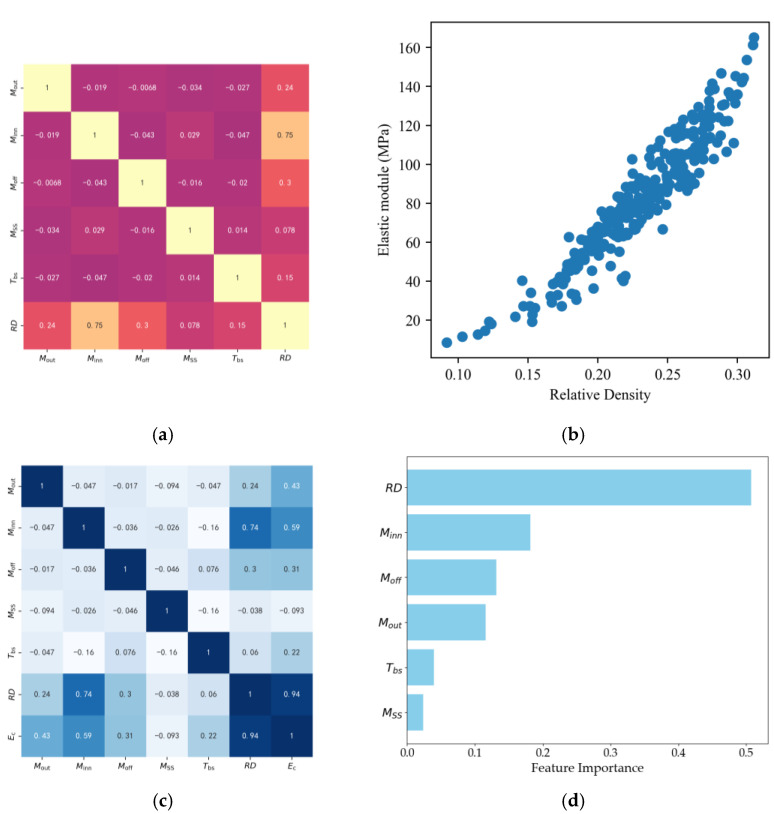
Sample space and Feature index: (**a**) Correlation analysis between *RD* and geometric parameters. (**b**) Data distribution. (**c**) Correlation analysis between *RD*, geometric parameters and Young’s modulus. (**d**) Feature importance of predicting $E_{c}$.

**Figure 10 materials-17-05599-f010:**
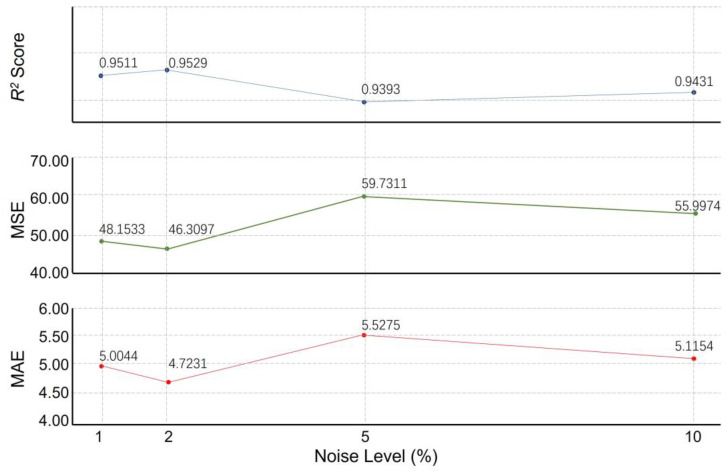
Noise level analysis.

**Figure 11 materials-17-05599-f011:**
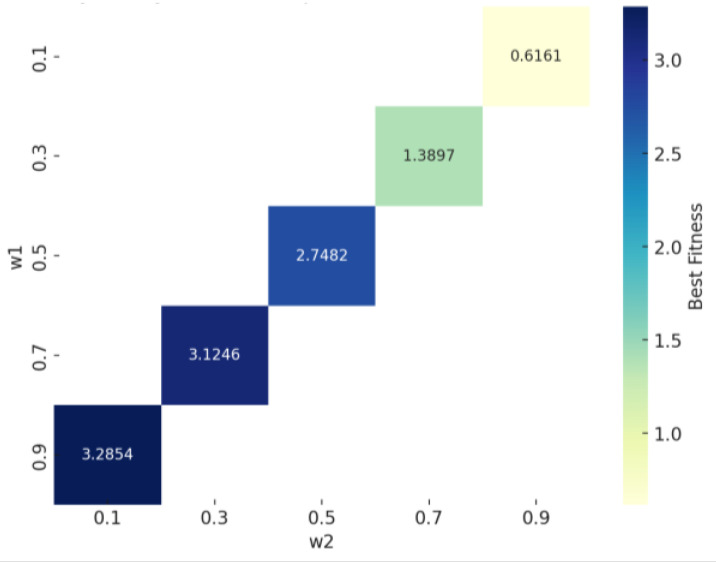
$w_{1}$ and $w_{2}$ sensitivity analysis result.

**Figure 12 materials-17-05599-f012:**
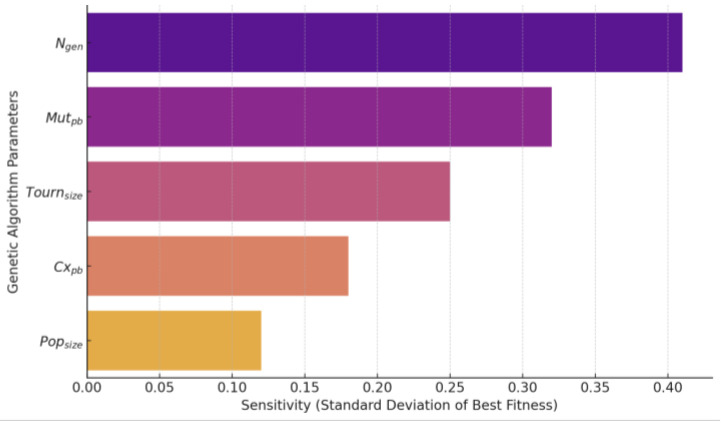
Genetic algorithm parameter sensitivity analysis.

**Figure 13 materials-17-05599-f013:**
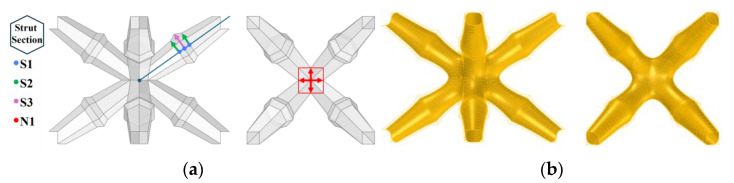
Parametric lattice design based on subdivision modeling: (**a**) design parameters and mesh model; (**b**) SubD model of lattice model.

**Figure 14 materials-17-05599-f014:**
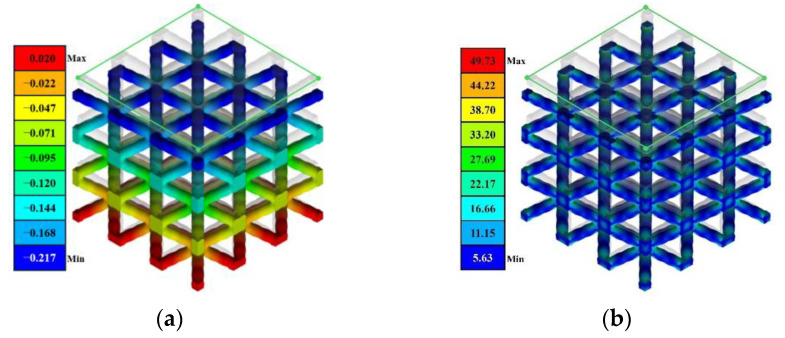
Mechanical response of RVE under a tensile load. (**a**) Uniaxial deformation in *Z*-axis; (**b**) von Mises stress distribution.

**Figure 15 materials-17-05599-f015:**
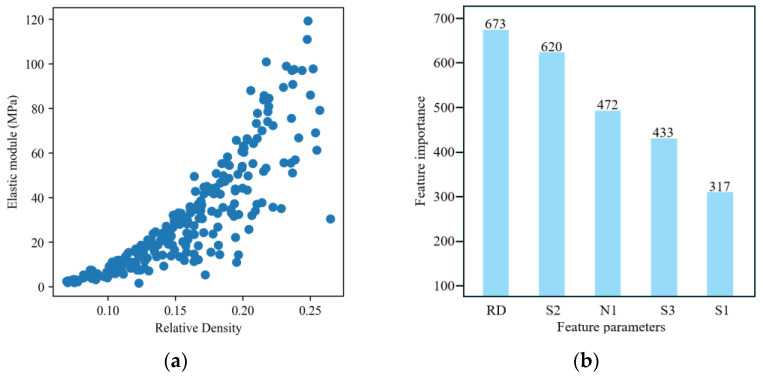
Sample space and Feature index: (**a**) Data distribution. (**b**) Feature importance.

**Table 1 materials-17-05599-t001:** Parametric design domain.

Parameter	$M_{inn}$	$M_{out}$	$M_{strut}$	$M_{smooth}$	$T_{size}$
Value range (mm)	[0.30–0.50]	[0.30–0.50]	[0.5–1.5]	[1.0–10.0]	[1.0–1.5]
Step	0.01	0.01	0.1	0.1	0.1
Step number	21	21	11	101	6

**Table 2 materials-17-05599-t002:** MAE and R^2^ of the employed ML model.

	CV MAE	CV RMES	CV R^2^ Mean	CV R^2^ Std Deviation
Polynomial Regression	0.0073	0.011	0.829	0.065
Random Forest Regression	4.509	6.017	0.965	0.010

**Table 3 materials-17-05599-t003:** $w_{1}$ and $w_{2}$ weight parameter sampling.

Weight Factors	Factor Combinations				
$w_{1}$	0.1	0.3	0.5	0.7	0.9
$w_{2}$	0.9	0.7	0.5	0.3	0.1

**Table 4 materials-17-05599-t004:** Experiment validation.

$E_{c}^{\mathrm{target}}$ (MPa)	80	100	120	140
Unit models	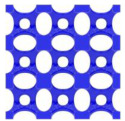	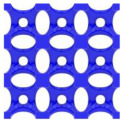	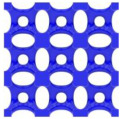	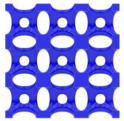
$M_{inn}$	0.50	0.40	0.49	0.47
$M_{out}$	0.31	0.44	0.40	0.46
$M_{smooth}$	4.88	7.4	3.66	7.0
$M_{strut}$	1.1	0.88	1.35	0.75
$T_{size}$	1.3	1.32	1.38	1.41
$E_{c}^{\mathrm{sim}}$ (MPa)	80.6	108.5	124.5	151.7
$E_{c}^{\exp}$ (MPa)	16.73	17.49	18.27	23.2
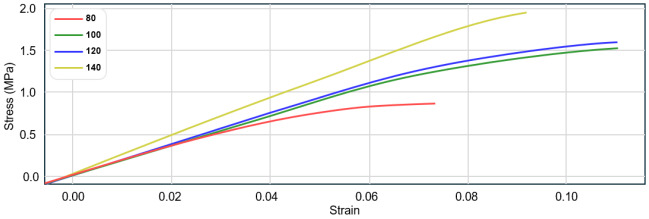				

**Table 5 materials-17-05599-t005:** Predicted result and actual result in relative density elastic modulus.

	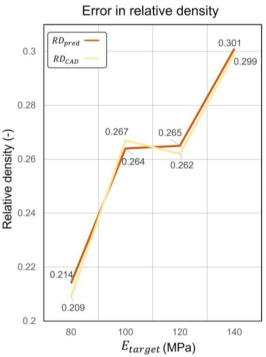			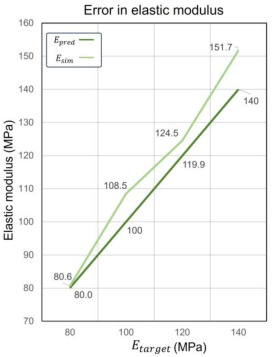		
$E_{\mathrm{target}}$ **(MPa)**	${\mathrm{RD}}_{\mathrm{pred}}$	${\mathrm{RD}}_{\mathrm{CAD}}$	**Error (*RD*)**	$E_{\mathrm{pred}}$ **(MPa)**	$E_{\mathrm{sim}}$ **(MPa)**	**Error (E)**
80	0.214	0.209	2.25%	80	80.6	0.75%
100	0.264	0.267	−1.18%	100	108.5	8.55%
120	0.265	0.262	1.24%	119.9	124.5	3.75%
140	0.301	0.299	0.49%	140	151.7	8.38%

**Table 6 materials-17-05599-t006:** Parametric design domain.

Parameter	S1	S2	S3	N1
Value range (mm)	[0.35–0.45]	[0.20–0.90]	[0.20–0.90]	[0.50–1.20]
Interval (mm)	0.1	0.2	0.2	0.2

**Table 7 materials-17-05599-t007:** MAE and R^2^ of the employed ML model.

	CV MAE	CV R^2^
Polynomial Regression	0.0049	0.98187
Random Forest Regression	4.5046	0.91912

**Table 8 materials-17-05599-t008:** Experiment validation.

ID	BCC	C1	C2	C3	C4
Lattice unit	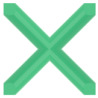	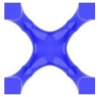	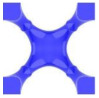	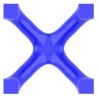	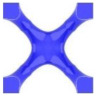
As-fabricated model	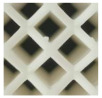	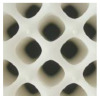	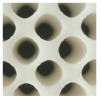	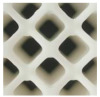	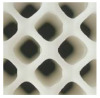
S1	-	0.44	0.46	0.43	0.45
S2	-	0.44	0.36	0.58	0.48
S3	-	0.60	0.62	0.62	0.60
N1	-	0.92	0.98	0.76	0.86
*RD*	0.156	0.160	0.153	0.155	0.156
Es (MPa)	8.97 (0%)	11.27 (25.6%)	10.34 (15.3%)	9.40 (4.8%)	10.90 (21.5%)
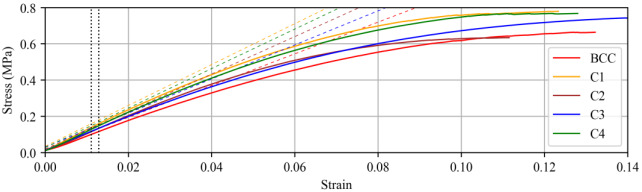

## Data Availability

The original contributions presented in the study are included in the article, further inquiries can be directed to the corresponding author.

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
