# Peer review of "Data-Driven Bi-Directional Lattice Property Customization and Optimization"

_materials, 2024, doi:10.3390/ma17225599_

Round 1

Reviewer 1 Report

Comments and Suggestions for Authors

The Review Report is in the attachment.

Comments on the Quality of English Language

The English could be improved to more clearly express the research.

Author Response

Comments 1: Can the authors provide a more detailed explanation of the Sub-D modelling process, including the specific parameters used in the Catmull-Clark algorithm?

Response 1: We thank the reviewers for their interest in our Sub-D modelling process.

We have added a detailed textual and graphical description of the Sub-D modelling process to the revised manuscript, including specific details of how we used Rhino's SubD tools to create and optimize the model. This will help readers gain a deeper understanding of our modelling process and technological choices. In addition, since Rhino's SubD tool automatically sets the parameters of the Catmull-Clark subdivision algorithm, we are unable to provide specific algorithmic parameter settings. However, by carefully designing the control mesh, we are able to control the shape and features of the subdivision surfaces to meet the needs of the lattice design. The revised content is seen in Page 3, Lines 129-151.

Comments 2: How did the authors determine the weighting factors (w1 and w2) in the objective function (Equation 12)? Was any sensitivity analysis performed on these weights?

Response 2: Thank you for your insightful question regarding the determination of the weighting factors w1 and w2​ in our optimization objective function.

The weighting factors and ​ are crucial in adjusting the trade-off between elastic modulus and relative density during the optimization. We determined these weights based on sensitivity analysis, as seen in Page 13, Lines 416-429.

Comments 3: While the authors mention using polynomial regression and Random Forest, important details are missing. What were the hyperparameters of the Random Forest model? How was the model trained and validated (e.g., train-test split, cross-validation)? A more detailed description of the ML model selection and training process is necessary. Did the authors explore other machine learning algorithms (e.g., support vector regression, neural networks)?

Response 3: Thanks for your interest and suggestions. We recognize the importance of providing more detailed information on machine learning models to enhance the transparency and reproducibility of our research.

In this study, we used the Polynomial Regressor and Random Forest Regressor model from the scikit-learn library. To optimize the model performance, we performed hyperparameter tuning, used Grid Search and 5-fold cross-validation. Please see Page 11, Lines 364-382 for details.

Response for “Did the authors explore other machine learning algorithms (e.g., support vector regression, neural networks)?”

We mainly used polynomial regression and random forest models based on the following considerations:

First, these models have good interpretability and simplicity. Polynomial regression can capture the nonlinear relationship between variables in an intuitive way, which helps us understand the effect of each input geometric parameter on the output. Random Forest performs well in dealing with complex nonlinear interactions, is highly resistant to overfitting, and provides a measure of feature importance.

We did not use support vector regression and neural networks, mainly because these methods typically require larger datasets and more computational resources. Given the size of our dataset and the time constraints of our study, we chose models that strike a balance between computational efficiency and predictive performance. In addition, interpretability is critical to our research, and neural networks and support vector regression are often viewed as ‘black box’ models that make it difficult to intuitively interpret the relationship between inputs and outputs.

We acknowledge that exploring other machine learning algorithms may help improve the predictive accuracy and generalization of the models. In future research, we plan to experiment with these methods, evaluate their performance on our dataset, and consider integrating multiple models to further improve prediction.

Comments 4: Similarly, details about the GA parameters (population size, mutation rate, crossover rate, number of generations, etc.) are lacking. How were these parameters chosen? Sensitivity analysis of these parameters would strengthen the study.

Response 4: Thank you for your valuable advice. We are aware of the impact of parameter settings on the performance of genetic algorithms. To balance the exploratory and exploitative capabilities of the algorithm to achieve satisfactory optimization results within a reasonable computational time, we conducted a parameter sensitivity analysis, focusing on the impact of factors such as population size, mutation rate and crossover rate on the optimization results. We have reported the results of the sensitivity analyses in detail in the revised manuscript, including comparisons of performance under different parameter settings, to increase the rigour of the study. Please see Page 13, Lines 424–455.

Comments 5: How robust is the two-tiered ML model to noise in the input data? Did the authors perform any analysis to assess the model's sensitivity to noise?

Response 5: Thank you for your question regarding the robustness of our two-tiered machine learning model to noise in the input data. We conducted a sensitivity analysis to assess how noise affects model performance.

We introduced Gaussian noise into the input features at varying levels (1%, 2%, 5%, and 10% of each feature's standard deviation) to simulate measurement errors. For each noise level, we retrained the polynomial regression and Random Forest models and evaluated them on the original test set.

Our findings showed that the Random Forest model is particularly robust for input noise, with only a slight decrease in performance even at higher noise levels. The polynomial regression model exhibited more sensitivity to noise but maintained reasonable accuracy. Please see Page 12, Lines 384-399.

Comments 6: The authors should provide a more detailed discussion of the convergence criteria used for the genetic algorithm.

Response 6: Thank you for your question about the convergence criteria used in our genetic algorithm. We appreciate the opportunity to clarify this aspect of our methodology.

In our study, the convergence of the genetic algorithm was determined by a sensitivity analysis to ensure efficient and effective optimization. First, we set a maximum number of generations at 20, determined by preliminary sensitivity analysis where we observed that significant fitness function improvements typically occurred within the first 15-20 generations. This limit helped to avoid unnecessary computation when further iterations showed diminishing returns. We have discussed the convergence performance of the algorithm in more detail in the revised paper. Please also see Page 12, Lines 384-399.

Comments 7: The authors should state how does the proposed method compares to other state of-the-art methods for lattice structure optimization in terms of accuracy, efficiency, and design space exploration?

Response 7: Thank you for your question about how our proposed method compares to other state-of-the-art methods for lattice structure optimization in terms of accuracy, efficiency, and design space exploration.

Our method offers several advantages over traditional approaches. In terms of accuracy, by employing a two-tiered machine learning framework that incorporates relative density as a key input feature, we achieve more precise predictions of mechanical properties. This allows for better control over target properties compared to methods that rely solely on geometric parameters.

Regarding efficiency, integrating generative design with machine learning significantly reduces computational costs. Unlike conventional methods that require extensive finite element analysis or iterative simulations, our approach streamlines the process by training predictive models, enabling us to achieve target properties with fewer design iterations. This accelerates the optimization process and makes it feasible to handle larger datasets and explore more complex design variations.

In terms of design space exploration, our method's bidirectional capability supports both forward and inverse design, allowing for broader exploration by generating structures to meet specific property targets and discovering new design configurations that may be overlooked by conventional methods. This flexibility provides a more comprehensive solution for tailoring lattice structures to meet a wide range of application-specific requirements.

Please see Page 20, Lines 654-686 for revised content.

Comments 8: The authors should improve the discussion section, and they should address more thoroughly the limitations of their study, including the assumptions made, potential sources of error, and the scope of applicability of the proposed method.

Response 8: Thank you for your insightful feedback. We appreciate the opportunity to enhance our discussion section by thoroughly addressing the limitations of our study, including the assumptions made, potential sources of error, and the scope of applicability of our proposed method. Please see Page 19, Lines 598–562 for revised content.

Comments 9: Fig. 1) lacks sufficient detail and clarity. Improving the figure caption will enhance readability.

Response 9: Thank you for bringing to our attention that Figure 1 lacked sufficient detail and clarity. We appreciate your suggestion to improve the figure caption to enhance readability. In response, we have revised both the figure and its caption to provide a clearer and more detailed explanation of our proposed methodology. Please see Page 3, Lines 110-127.

Reviewer 2 Report

Comments and Suggestions for Authors

In the manuscript "Data-driven bi-directional lattice property customization and optimization" the authors proposed a method for lattice design based on machine learning, including reverse design of crystal lattice.  The results obtained within the study may contribute to a better understanding of lattice morphology and skeleton parameters. In addition, the presented methodology may help predict the properties of crystalline materials and even design new materials with specific properties. I believe that the subject of the manuscript is relevant and that the results may be beneficial for the materials design community. Hence, I recommend the publication of the manuscript after minor corrections and clarifications:
1. For a very long time, computational chemistry methods like the Density Functional Theory (DFT) approach were primarily used for predicting the properties of crystal lattices. Are there any advantages of the proposed methodology compared to the prediction of crystalline properties based on solid-state DFT calculations? The authors should briefly elaborate on this.
2. Authors have stated that they used 80% of the data for training and 20% for validation. Was there any criterion based on which authors selected which data will be used for data training, and which for validation?
3. Authors have stated that "ultimately 3 lattices were selected from each group for compression testing". How were these structures chosen over others? Were all other lattices damaged and for that reason ignored, or is there another reason?

Author Response

Comments 1: For a very long time, computational chemistry methods like the Density Functional Theory (DFT) approach were primarily used for predicting the properties of crystal lattices. Are there any advantages of the proposed methodology compared to the prediction of crystalline properties based on solid-state DFT calculations? The authors should briefly elaborate on this.

Response 1: Even though DFT is still a crucial technique for comprehending and forecasting material properties at the quantum level, its applicability to mesoscale, complex lattice structures is frequently constrained by computational demands and scalability issues. In contrast, our proposed data-driven bi-directional approach is specifically designed for mesoscale to macroscale applications, especially when it comes to tackling complicated geometries and multi-objective optimization problems, our proposed data-driven bi-directional approach provides notable benefits in terms of computing efficiency, inverse design abilities, and flexibility. We can quickly predict mechanical properties and optimize lattice materials using machine learning and genetic algorithms. This makes our methodology a useful addition to conventional DFT methodologies, particularly in applications that call for specialized material solutions and accelerated design cycles.

Comments 2: Authors have stated that they used 80% of the data for training and 20% for validation. Was there any criterion based on which authors selected which data will be used for data training, and which for validation?

Response 2: Thank you for your question regarding the criteria used to select data for training and validation in our machine learning (ML) process.

To divide the dataset into a training set and a validation set, we used a random splitting approach, shuffling the entire dataset and then assigning 80% of the data to the training set and 20% to the validation set. This random shuffling is crucial to avoid possible biases in the order of the raw data, such as temporal or sequential patterns that may affect the training process. The selection was completely random to ensure that both subsets were representative samples of the complete dataset, keeping the distribution of features and target variables consistent. This approach facilitates an unbiased assessment of the model's performance and improves its generalizability to unseen data.

We have updated the manuscript to clarify this point and added a description of the random shuffling and splitting procedure in Page 11, Lines 360-363.

“In the ML training process, the dataset was split into a training set and a validation set (80% of the data for training and the remaining 20% for validation) using a random split technique. To prevent bias in the sequence of the data for model training, the data was randomly shuffled before splitting.”

In addition, we introduce cross-validation to ensure the reliability of the model's predictive performance. We believe that this addition improves the transparency of the model training process.

Comments 3: Authors have stated that "ultimately 3 lattices were selected from each group for compression testing". How were these structures chosen over others? Were all other lattices damaged and for that reason ignored, or is there another reason?

Response 3: Thank you for your careful review of our experimental process.

The selection of three lattice structures from each group for compression testing was based on practical and structural criteria. Some lattices were compromised by printing defects, such as delamination issues from our Formlabs equipment, or were damaged during support removal, making them unsuitable for testing. We chose three intact lattices per group to ensure geometric and structural integrity and maintain statistical robustness, balancing representativeness with experimental feasibility. This selection allowed us to assess manufacturing reproducibility and consistency of results.

We recognize that the original manuscript lacked detail in describing the sample selection process, which may have raised questions. The revised manuscript now includes a clear explanation of the selection criteria and process to enhance transparency and reproducibility. Please see Page 14, Lines 462-471.

Round 2

Reviewer 1 Report

Comments and Suggestions for Authors

The authors responded correctly to the reviewer's comment and he found the revision satisfactory. The manuscript could be accepted for publication in Materials.